# Towards Non-Invasive and Continuous Blood Pressure Monitoring in Neonatal Intensive Care Using Artificial Intelligence: A Narrative Review

**DOI:** 10.3390/healthcare11243107

**Published:** 2023-12-06

**Authors:** Stephanie Baker, Thiviya Yogavijayan, Yogavijayan Kandasamy

**Affiliations:** 1College of Science and Engineering, James Cook University, Cairns, QLD 4878, Australia; 2College of Medicine and Dentistry, James Cook University, Townsville, QLD 4811, Australia; thiviya.yogavijayan@my.jcu.edu.au; 3Department of Neonatology, Townsville University Hospital, Townsville, QLD 4811, Australia; yoga.kandasamy@health.qld.gov.au

**Keywords:** artificial intelligence, blood pressure, neonatal medicine

## Abstract

Preterm birth is a live birth that occurs before 37 completed weeks of pregnancy. Approximately 11% of babies are born preterm annually worldwide. Blood pressure (BP) monitoring is essential for managing the haemodynamic stability of preterm infants and impacts outcomes. However, current methods have many limitations associated, including invasive measurement, inaccuracies, and infection risk. In this narrative review, we find that artificial intelligence (AI) is a promising tool for the continuous measurement of BP in a neonatal cohort, based on data obtained from non-invasive sensors. Our findings highlight key sensing technologies, AI techniques, and model assessment metrics for BP sensing in the neonatal cohort. Moreover, our findings show that non-invasive BP monitoring leveraging AI has shown promise in adult cohorts but has not been broadly explored for neonatal cohorts. We conclude that there is a significant research opportunity in developing an innovative approach to provide a non-invasive alternative to existing continuous BP monitoring methods, which has the potential to improve outcomes for premature babies.

## 1. Introduction

Worldwide, an estimated 15 million neonates are born preterm each year, accounting for nearly 1 million deaths yearly [1]. In Australia, more than 26,000 neonates are born preterm each year [2]. Whilst the survival for premature neonates has improved over the years, prematurity is still associated significantly with short-term and long-term complications [3]. The main cause of morbidity and mortality is organ immaturity. One commonly seen problem in the immediate extra uterine life is the difficulty in maintaining blood pressure (BP), which reflects the immature cardiovascular system. Blood pressure reading is reported as systolic (SBP) and diastolic blood pressures (DBP). From these measurements, mean arterial pressure (MAP) is calculated by MAP = DBP + $\frac{1}{3}$(SBP − DBP) [4].

BP monitoring is essential for managing haemodynamic instability in premature and critically ill neonates [5]. Approximately 20% of premature neonates develop hypotension in the first 24 h of life [6]. There is a myriad of factors that contribute to low blood pressure in a premature neonate. Blood loss during birth, increased endothelial permeability, immature heart with ineffective cardiac output, the presence of a patent ductus arteriosus, exposure to antenatal steroids, delayed cord clamping, and use mechanical ventilation are among the factors that influence blood pressure [7]. Hypotension is associated with double the risk of intraventricular haemorrhage, which can result in severe long-term neurological consequences in those who survive this complication [8]. 

The Haemodynamics Working Group of the International Neonatal Committee has published a review addressing methods of BP measurement in neonates, highlighting the key challenges in obtaining reliable BP measurements [5,9]. The primary methods for measuring BP in neonates are cuff-based infant sphygmomanometers and continuous arterial BP monitoring, as illustrated in Figure 1. Intra-arterial monitoring, either with an umbilical artery or peripherally inserted arterial line is highly invasive and introduces the risk of significant complications such as thrombosis and infection in vulnerable neonates [10]. This methods is also limited to clinical settings performed by skilled clinicians, making it unsuitable for use in low-resource environments [11]. 

Meanwhile, the widely accepted method of non-invasive BP measurement is the sphygmomanometer, an oscillometric method that involves an inflatable cuff generally placed on the upper arm. The cuff is inflated to a pressure beyond the expected systolic BP and then slowly reduce. The maximum and minimum cuff pressures at which blood flow turbulence is audible to inbuilt sensors or via a stethoscope are taken as the systolic and diastolic BP readings, respectively. While this method is non-invasive, the application of high pressures can cause discomfort. It is also only capable of measuring BP intermittently, preventing continuous monitoring. Additionally, issues including low accuracy and user error [9,12] have previously been identified. 

The limitations of oscillometric measurements have led to researchers looking at alternative methods for BP measurement that can be used continuously. Several recent studies have sought to utilize artificial intelligence (AI) algorithms to estimate BP from heart activity waveforms obtained using sensors such as photoplethysmogram (PPG) and electrocardiogram (ECG) [13,14,15,16,17,18]. These sensors are routinely used in critical care settings for cardiorespiratory health monitoring purposes, however, have not yet been validated for monitoring BP. Several studies have investigated various AI approaches towards measuring BP from heart activity signals in adult cohorts, showing some promise that has not yet been translated into successful clinical tools for widespread use. In addition, neonatal cohorts have not been broadly considered. Thus, there is a significant gap in the literature pertaining to non-invasive BP measurement in this group that could lead to a significant change in clinical practice.

This narrative review contributes to the literature by analysing the state-of-the-art literature on non-invasive and continuous blood pressure monitoring, with a focus on the neonatal cohort. We identify that there are very few studies investigating this topic for neonates; however, a substantial body of literature exists for adult cohorts. We then explore key technologies, including non-invasive sensors and artificial intelligence, which are critical to the development of non-invasive BP monitoring tools. Additionally, we introduce key metrics for assessing the performance of BP monitoring tools. As most identified works are focused on adult cohorts, we therefore place a lens on the few pioneering works which have considered the neonatal cohort. We ultimately discover a significant need for further research in this field, and as such, conclude this paper by highlighting directions that offer significant opportunity for future researchers. 

## 2. Materials and Methods

Our narrative review was conducted by first searching PubMed, Scopus, and Web of Science for relevant studies. Our keywords included “non-invasive”, “cuffless”, “wearable”, “blood pressure”, “prematur*”, “preterm”, “neonat*”, “baby”, “infant”, “pediatric”, “paediatric”, “child*”, “artificial intelligence”, “neural network”, “deep learning”, and “machine learning”. Studies that focused on the non-invasive measurement of blood pressure or the use of AI to predict blood pressure in any age cohort were considered, with papers focused on the neonatal cohort included by default. Non-English papers were also excluded. As the field of artificial intelligence is rapidly evolving with ever-improving computational resources, we focused our search on papers published between 2017 and 2023. Due to the large volume of papers in the adult cohort, we selected only papers published in reputable journals that utilised technologies which may be suitable for the neonatal cohort.

The first outcome of this literature search was to identify several non-invasive sensors that can be used to obtain cardiac activity waveforms on a continuous basis. The second key outcome was to investigate the suitability of various artificial intelligence tools for developing a predictive algorithm for estimating blood pressure, using metrics obtained from non-invasive devices. 

## 3. Results

Our exploratory search revealed that research on non-invasive BP measurement for neonatal patients, particularly those born very preterm, is limited. However, a larger body of literature investigates non-invasive BP measurement for paediatric and adult cohorts. In this section, we explore candidate sensors, artificial intelligence approaches, and assessment metrics for non-invasive blood pressure monitoring devices. As these findings are primarily based on research in adult cohorts, we present a short section that outlines the limited research conducted in neonatal and paediatric cohorts, before summarizing the potential clinical outcomes that these techniques offer to critical care monitoring for preterm infants.

### 3.1. Architecture for Non-Invasive Blood Pressure Monitoring

In this subsection, we highlight the key sensor and AI technologies utilized for developing non-invasive BP monitoring systems in the literature, as illustrated in Figure 2. Additionally, the key metrics used to assess the performance of these systems are discussed. Most of the discussed works have focused on an adult cohort; however, similar techniques could be applied in a neonatal cohort in future works.

#### 3.1.1. Sensors

A range of non-invasive sensors have been explored in the literature for the non-invasive monitoring of BP parameters in adult cohorts, including photoplethysmogram (PPG) [13,15,16,17,19], electrocardiogram (ECG) [13,14,18], bioimpedance sensors [20,21], pressure sensors [22], remote PPG (rPPG) [16], and ultrasonic transducers [23]. Of these, only PPG and ECG have been explored for neonatal cohorts directly [24]. Several of these devices are currently used in clinical practice for the measurement of other health parameters. For example, PPG is used for monitoring both blood oxygen saturation and heart rate. However, none of these sensors have been broadly validated or approved for use in monitoring BP in clinical settings.

PPG and ECG are particularly prevalent in the literature, likely due to their widespread use in clinical practice for cardiorespiratory health monitoring purposes. PPG sensors are routinely used in a small finger-clip device to monitor heart rate and blood oxygen saturation, while ECGs are regularly used to monitor heart rate and rhythm. As such, a validated BP monitoring technique for BP monitoring based on one or both sensors would be straightforward to implement in clinical settings given that the hardware is already available and would carry the benefit of needing no additional sensors to be applied to vulnerable neonatal skin. 

Data obtained via PPG and ECG sensing has been used in a variety of ways in the literature. Many early works sought to extract pulse transit time (PTT), typically measured as the time distance between ECG R-peak and PPG peak, and correlate this with blood pressure metrics using AI techniques or other approaches [25,26,27]. While the results of these early works showed that PTT was a reasonable proxy for BP, the need for the ongoing time synchronization of two separately measured signals is a substantial challenge in hardware design that is difficult to overcome without regular calibration.

With increases in computational power and advancements in AI, an increasing body of work has instead used short segments of PPG or ECG waveforms as inputs to AI models [13,15,16,17,18], with promising results. Several other works have also investigated the development of more computationally efficient methods, including extracting features that describe the shape of PPG and ECG waveforms [14,28] and extracting non-linear features of PPG waveforms [19] as inputs for AI models. The lower computational requirements of these methods may make them more suitable for wireless battery-powered devices.

Aside from wearable PPG- and ECG-based approaches, there are several alternative and novel sensing approaches that also carry promise for the neonatal cohort. While PPG is typically obtained via a wrist- or finger-based wearable device, one recent study [16] instead extracted PPG waveforms from RGB camera imagery by analysing changes in skin tone due to heart activity before using these signals to train a hybrid neural network (NN) to predict SBP and DBP. The reported error was significant and thus currently unsuitable for clinical use; however, the approach is interesting given it is entirely non-contact. Further investigation may be warranted to improve the accuracy of this method.

One work focused on a neonatal cohort used a capacitive pressure sensing array [29] built into a wearable wrist band. The capacitive sensor array was designed to measure displacement with high sensitivity. Two electrodes were built into a soft wearable band, with one electrode designed to sit against the wrist and move with the pulse while the other electrode remains relatively fixed. The small displacement changes caused by the pulse are measured as a change in capacitance, which in turn enables pulse waveforms to be derived. The waveforms obtained using this sensing method were then utilized to train a neural network to predict BP parameters, showing promising early results on a small cohort based on standard deviation. As this approach is not optical, it is likely to be unaffected by skin tone, thus it is likely to work equally well on a broad range of infants; however, this would need to be validated. A potential disadvantage is that its high sensitivity to displacement may lead to increased motion artifacts compared to other methods. This sensing method offers a promising direction for future research. 

In another recent work by Si et al. [22], a textile-based pressure sensing approach was used to measure ballistocardiogram (BCG) signals, which capture vibrations of the body caused by heart activity. Using nylon as a substrate, they embedded a flexible pressure sensing array into a flexible chair covering. Through feature importance analysis, they identified several key features of the BCG waveform that were highly correlated with diastolic or systolic BP. Thereafter they trained a neural network (NN) to predict BP from BCG features, achieving promising results. While their study focused on an adult cohort, this sensing strategy could potentially be suitable for incorporation into bed sheets for neonatal care units. BCG sensors have also previously been utilised in the neonatal cohort for other applications, including motion detection [30], and as such, BP sensing could be combined with other essential monitoring requirements using BCG.

Bio-impedance sensing arrays have also been considered by several recent papers [20,21]. These sensors capture pulsatile activity by measuring electrical impedance across an area of interest, commonly the wrist. They have primarily been suggested as an alternative to PPG, largely due to bioimpedance sensors achieving more consistent performance across diverse skin tones than PPG sensors [21]. Limited results were reported in either paper; however, the standard deviation values suggest reasonable performance. 

Overall, a broad range of sensors have been considered for blood pressure monitoring in adult cohorts, however only PPG and ECG have been considered in neonatal cohorts. PPG is a particularly appealing approach given that it is already routinely used for continuous monitoring in neonatal critical care settings, and as such, the development of a validated BP measurement scheme based on PPG could be rapidly deployed through a software update. However, approaches discussed in this section offer advantages such as lower contact (e.g., rPPG) and more consistent performance in diverse cohorts (e.g., capacitive sensing); as such, these alternatives warrant further exploration and efforts to improve affordability.

#### 3.1.2. Artificial Intelligence

Artificial intelligence (AI), or machine learning (ML), has proven critical in calculating BP parameters, including systolic blood pressure (SBP), diastolic blood pressure (DBP), and mean arterial pressure (MAP), from the waveforms and other data obtained via non-invasive monitoring. In the literature, AI approaches have been applied to both raw waveform data [13,15,16,17,18] and hand-picked features extracted from waveform data [14,19,20,22]. AI methods used in this space are varied; however, most recent works have focused on neural network (NN) architectures [13,14,15,16,17,22,28].

Several early works considered various regression methods, including the work by Chung et al. [24] which used linear regression to link pulse transit time parameters derived from PPG, ECG, and seismograph (SCG) sensors with systolic BP in a neonatal cohort, showing promising results on a small dataset (n = 5). Works focused on adult cohorts also considered a range of regression methods, including linear regression [25], support vector machine [15,19,25], decision tree [15,25], and random forest [15,25]. However, the results obtained by these foundational models have been largely outperformed by neural networks, leading to the most recent literature instead focusing on various neural network architectures.

Fully connected neural networks (FCNNs) are the most fundamental type of neural network and have been considered in several recent works. In one work focused on a neonatal cohort, an FCNN was used to predict BP parameters from PPG data. The standard deviation (SD) was reported as 6.6 mmHg and 7.9 mmHg, respectively. Mean absolute error (MAE) results were reported; however, several of the reported values were negative, indicating that these values are likely mean error (ME). In another work focused on an adult cohort [22], FCNN was used to predict BP from features of BCG waveforms in an adult cohort, achieving MAE values of 3.90 mmHg and 4.62 mmHg for DBP and SBP, respectively. Although FCNN architectures have shown promising results, they are computationally expensive to train and are prone to overfitting. As such, they are less prevalent in the literature than other more advanced NN architectures. 

Convolutional neural networks (CNNs) and long short-term memory (LSTM) networks have been broadly explored for interpreting waveform data. CNN models perform strongly in identifying the key features of waveforms, while LSTM is highly suited for identifying relationships between key features, making them especially useful for interpreting time-series data. While several studies have considered these models on their own [16,20,31], most recent works have utilized hybridized CNN-LSTM model architectures to gain the benefits of both model types within a single structure.

One such work [32] utilized CNN and LSTM models in parallel, with the results then flattened into a single layer for the prediction of systolic and diastolic BP in an adult cohort. Two CNNs were trained to interpret the morphological and frequency spectrum characteristics of PPG waveforms, and one LSTM was trained to interpret temporal features of PPG waveforms, with all waveforms derived from a subset of MIMIC-II. Their results achieved ”B” and ”A” grades in accordance with the BHS Protocol for SBP and DBP, respectively. Additionally, MAE ± SD values of 5.59 ± 7.25 mmHg and 3.36 ± 4.48 mmHg were achieved for SBP and DBP, respectively, with the AAMI Standard thus met for DBP only. While SBP performance was slightly below industry standards, this approach shows promise in assessing waveform features in several ways. 

Sequential CNN-LSTM models were also considered in several works in adult cohorts [13,14,16], with these models featuring CNN layers that identify key features feeding into LSTM layers that identify the relationships between those features. In one recent work [13], the CNN-LSTM model shown in Figure 2 was used to predict systolic and diastolic BP from PPG and ECG waveforms derived from the adult cohort within Medical Information Mart for Intensive Care (MIMIC-III) database, achieving an “A” grade in accordance with the BHS Protocol and a “Pass” grade in accordance with the AAMI standard. They reported MAE ± SD values of 4.41 ± 6.11 mmHg and 2.91 ± 4.23 mmHg for SBP and DBP, respectively. Another work [16] utilized a CNN-LSTM model fine-tuned to individual participants to predict BP parameters in an adult cohort using PPG and rPPG signals, comparing this model to several deep CNN architectures from the literature. Using PPG data, they achieved MAE values of 9.0 mmHg and 4.6 mmHg for SBP and DBP, respectively. Using rPPG data, MAE values of 13.6 mmHg and 10.3 mmHg were reported for systolic and diastolic BP, respectively. These results were stronger than those of the other considered CNN models, but do not meet the AAMI Standard. Meanwhile, one work [14] used a shallow CNN-LSTM model to predict BP parameters based on 12 features describing the shape of PPG and ECG waveforms from the MIMIC-III database, with the results achieving “A” and “Pass” grades for the BHS Protocol and AAMI standard, respectively. The reported MAE ± SD values were 4.53 ± 6.27 mmHg and 3.37 ± 4.84 mmHg for systolic and diastolic BP, respectively. Additionally, explainable artificial techniques were used to identify the features most strongly linked with BP. This analysis showed that the ECG Q-wave and R-wave and PPG trough were most strongly predictive of SBP, while the PPG peak and trough and ECG R-wave were most strongly predictive of DBP.

Several recent works have also looked to improve on the performance of LSTM structures using attention mechanisms, which enable the model to learn contextual information about the input features. One such work [28] developed a simplified LSTM variant known as a gated recurrent unit model with an attention mechanism to predict diastolic and systolic BP from non-linear features of PPG waveforms, using a subset of 500 waveforms from the MIMIC-II database comprised of an adult cohort. Their proposed model achieved MAE ± SD values of 2.58 ± 3.35 mmHg and 1.26 ± 1.63 mmHg for SBP and DBP, respectively. These results meet the AAMI Standard and are shown to exceed results obtained via other models, including linear regression and vanilla LSTM. 

In another recent work, a novel Transformer architecture was considered for predicting BP parameters from PPG waveforms in an adult cohort [17]. Transformers are advanced AI models that have been gaining popularity in the literature due to their ability to capture global context and identify relationships across long time-series of data. Using 5 s segments of PPG waveforms from the MIMIC-III adult cohort, the transformer architecture proposed in [17] was used to develop both general and personalized models. The model trained on general data achieved MAE values of 4.97 ± 4.72 mmHg and 2.99 ± 2.39 mmHg for systolic and diastolic BP, respectively, while the model with personalized fine-tuning achieved MAE values of 2.41 ± 2.72 mmHg and 1.31 ± 1.77 mmHg, respectively. For both the general and personalized models, the AAMI Standard was satisfied, and the BHS Protocol grade of ‘A’ was achieved in measuring systolic and diastolic BP. These results are promising, showing great potential for this emerging class of artificial intelligence models.

Overall, a wide range of AI models have been explored for predicting BP parameters in adult cohorts. Most recent works have utilized neural networks, with advanced methods such as CNN-LSTM hybrids and transformer models showing particularly strong promise. In the neonatal cohort, these advanced AI techniques have not been explored. Works to date have focused on simpler methods, including linear regression and FCNN, with results showing room for improvement. As a result, there is a significant gap in the literature in applying advanced AI architectures that have shown promise in adult cohorts to the prediction of BP parameters in neonatal cohorts.

#### 3.1.3. Assessment Metrics

There are several approaches used in the literature to assess the performance of tools for BP monitoring. The first is straightforward error metrics, including mean absolute error (MAE), standard deviation (SD), and root-mean-square error (RMSE). MAE and SD provide insight into the magnitude and range of errors, while RMSE more heavily penalises high-range errors and thus helps to reveal whether the errors being made are significant or not. Several works in the literature report mean error (ME); however, this method does not adequately illustrate model performance due to negative and positive errors effectively cancelling either other out. While ME could be used as a first check that the model is not significantly underperforming, an ME close to zero does not provide any evidence that a model is performing well. 

In addition to numeric error metrics, tools including Bland–Altman plots and error histograms are broadly used. Bland–Altman plots show the level of agreement between a novel measurement technique and an existing gold-standard [33]. Most papers utilizing Bland–Altman plots compare their proposed technique with intra-arterial monitoring [13]; however, some compare to sphygmomanometer-based oscillometric measurement [22]. Error histograms can also be used to visualize the distribution of errors by splitting errors into small bins and plotting the frequency of errors in those bins. This has been used in several works [14,34]; however, it is less prevalent than Bland–Altman analysis. Nonetheless, error histograms are highly useful for understanding whether errors are evenly distributed between underestimation and overestimation error, as well as whether significant high-range errors are occurring in either direction.

Lastly, industry-based benchmarks are regularly used to assess tool performance, including the British Hypertension Society Protocol for the Evaluation of Blood Pressure Measuring Devices [35] (hereafter called the BHS Protocol) and the Association for the Advancement of Medical Instrumentation’s American National Standard for electronic or automated sphygmomanometers [36] (hereafter called the AAMI Standard). The BHS Protocol assigns a grade to blood pressure measurement devices based on the quantity of measurements meeting different error thresholds, as outlined in Table 1 below. Meanwhile, the AAMI Standard assigns “pass” grades to any device that achieves mean absolute error ≤ 5 mmHg and standard deviation ≤ 8 mmHg, while all others are awarded “fail”.

### 3.2. Non-Invasive Blood Pressure Monitoring in Neonatal Cohorts

Few works in the literature have investigated non-invasive blood pressure monitoring in neonatal cohorts. This subsection highlights the pioneering works in this domain, highlighting the need for further research in order to develop a clinically suitable method for non-invasive blood pressure monitoring in vulnerable neonatal cohorts.

One recent work by Chung et al. [24] developed a wireless wearable adhesive device containing ECG, PPG, seismocardiogram (SCG), and accelerometer sensors for use in a neonatal cohort. Their work first extracted a pulse arrival time (PAT) metric, measured as the time distance between ECG R-peaks and PPG troughs, as well as a pulse transit time (PTT) metric, measured as the time distance between ECG R-peaks and SCG troughs. Linear regression was then used to determine a relationship between systolic BP and PAT, as well as systolic BP and PTT. They reported SD values of 7.99 mmHg and 7.86 mmHg for PAT- and PTT-derived SBP values, respectively; however, MAE was not reported. Significant outliers were seen on their Bland–Altman plots; however, the authors report that this corresponded with significant movement detected on the accelerometer, suggesting that motion artifact led to errors in this period. Overall, initial results were promising; however, the cohort was small (n = 5), and several key statistics were not reported. As such, further validation of the proposed method is required. Additionally, diastolic BP and mean arterial pressure are not considered despite these parameters being of clinical significance. Finally, their proposed method depends on accurate time synchronization between the ECG and PPG waveforms. While steps were taken in their design process to ensure that this time synchronization is maintained, it is reported that some clock drift was seen over time; thus, the accuracy of the system is likely to decline without regular calibration.

In another work by Rao et al. [29], pulse waveforms were obtained using a capacitive sensor array proposed in an earlier work [37] that is capable of detecting the changing displacement of the ankle or wrist in response to the pulse. Their work then utilized neural networks in two stages of their design approach. Firstly, they trained a CNN to automatically assess the quality of the pulse waveform signals obtained from the device. This was then used to determine whether a signal was suitable for use in developing a blood pressure prediction model, with signals showing low-quality data excluded from the training and testing datasets. Thereafter, an FCNN was trained and tested for prediction of systolic, diastolic, and mean arterial BP. They reported ME ± SD values of −0.1 ± 7.9 mmHg, 0.1 ± 6.6 mmHg, and −0.1 ± 6.4 mmHg for systolic, diastolic, and mean arterial BP, respectively. While the ME values do not provide useful information as to the performance of the model or the distribution of errors, the SD values show some promise. Additionally, *r*^2^ regression scores of 0.64, 0.35, and 0.53 are reported for systolic, diastolic, and mean arterial BP, respectively. These results indicate that the FCNN model is not achieving a strong fit to the data, and as such, further research is certainly needed to improve upon these results. It is likely that more advanced AI models could achieve stronger results, such as CNN-LSTM and transformer architectures, which have performed well on adult cohorts.

Aside from the works by Chung et al. [24] and Rao and Quan et al. [29,37], there is relatively little research regarding the use of non-invasive sensing to monitor blood pressure in neonatal ICUs. Two further works by Revathi et al. [38] and Kapur et al. [39] considered blood pressure measurement in a broader paediatric cohort, including infants. The work by Revathi et al. [38] captured pulse activity in infants using a finger clip PPG sensor. They assessed the performance of several AI algorithms in measuring a range of vital sign parameters, reporting that support vector machine and random forest approaches showed the highest accuracy. However, specific results for BP measurement are not reported, so no conclusions can be drawn about the efficacy of these modes for BP monitoring specifically. In the work by Kapur et al. [39], an acoustic sensor was used to monitor heart sounds non-invasively in a paediatric cohort comprised of 25 patients aged 0–18 years. They reported RMSE values of 7.305 mmHg and 5.081 mmHg as well as ME ± SD values of 0.623 ± 7 mmHg and −0.051 ± 5 mmHg for systolic and diastolic BP, respectively. Additionally, the Pearson’s coefficient of correlation achieved were 0.964 and 0.935. Taken together, these results show promise for this method; however, it would require validation on a larger cohort and the analysis of relevant metrics such as MAE. Additionally, neonates have different normal ranges of blood pressure to older children, and thus, a model trained specifically for neonates is preferable in neonatal intensive care settings.

Overall, minimal research has been conducted on non-invasive and continuous blood pressure monitoring in the neonatal cohort. The studies that have been conducted in this space show promising initial results; however, they suffer from limitations, including small cohorts and falling short of metrics set by industry standards for BP measurement devices. There remains significant opportunity for future research in this space.

### 3.3. Limitations

This work presents a comprehensive narrative review of non-invasive and continuous blood pressure monitoring methods in the neonatal cohort. Every effort has been made to include all papers focused on this topic. Due to the relatively small body of work in this domain, our narrative review was expanded to include papers focused on the adult cohort which utilized technologies that are likely to be candidates for use in the neonatal cohort. The key limitation of this paper is that the review is narrative, not systematic, and thus, there may be additional papers in the adult cohort which are not included here despite being relevant to the topic. However, there are many reviews focusing on non-invasive blood pressure monitoring in other cohorts, for example, the recent works by Maqsood et al. [40] and Zhao et al. [41].

## 4. Opportunities for Translational Research

The field of non-invasive and continuous blood pressure monitoring for neonates remains in its infancy, with very few studies conducted in this domain. Given the importance of monitoring BP parameters in neonatal critical care and the limitations of existing techniques, there remains a significant need to develop a reliable and accurate tool for this task. This technology, if established, could be useful in neonatal intensive care settings or for neonates born outside a tertiary perinatal centre and subsequently requiring specialized aeromedical retrieval to a tertiary neonatal unit. Mortality rates remain higher for outborn (neonates born outside a tertiary perinatal centre) extremely premature neonates compared with their inborn peers [42], and this technology could help narrow the survival gap.

Research focused on the neonatal cohort have so far only considered limited data types, including PTT and PAT derived from PPG, ECG, and SCG signals [24], and pulse waveforms obtained from capacitive pressure sensing [29,37]. However, in the adult cohort, it has been shown that improved results can be obtained by using complete segments of raw PPG or ECG waveforms, indicating that alternative methods of data preprocessing can improve model performance. As PPG is affordable, broadly available, non-invasive, and routinely used in medical settings, we suggest that future studies should investigate the improved use of data obtained from this sensor. A validated approach using PPG sensors could be rapidly deployed in neonatal critical care units and would be suitable for use in low-resource settings. There also remains value in investigating novel sensor, including textile-embedded pressure sensing arrays and non-contact rPPG. These sensor types offer benefits to the neonatal cohort in terms of minimizing skin contact but have not yet been considered for BP monitoring in neonatal cohorts.

Advanced AI models have also not been broadly explored in neonatal cohorts, with studies to date using early models such as linear regression [24] and FCNN [29]. Given the strong performance of CNN-LSTM and transformer architectures in works investigating BP monitoring for adult cohorts, it is likely that these models would also perform well in neonatal cohorts. There remains significant research opportunity in applying advanced AI approaches to data from non-invasive sensors for the neonatal cohort.

Explainable artificial intelligence techniques have not been considered in the literature for neonatal BP monitoring. These techniques can help to reveal biomarkers that are strongly linked with BP parameters, which in turn can guide future research into treatments and medications to control BP levels. This biomarker identification approach has been considered for an adult cohort [14]; however, it remains an open challenge for the neonatal cohort.

Overall, this field remains open for significant future research, with challenges remaining across sensor technologies, data preprocessing techniques, artificial intelligence, and biomarker detection. This is an exciting field of research offering much opportunity to future researchers.

## 5. Conclusions

The use of artificial intelligence to predict blood pressure in premature neonates from non-invasive and continuous data has significant potential to improve clinical practice. Existing methods for BP measurement suffer from a myriad of issues, with gold-standard intra-arterial blood pressure monitoring being highly invasive and thus carrying various risks, while sphygmomanometers suffer from inaccuracies and an inability to be continuously used. 

In this work, we have conducted a scoping narrative review of the state-of-the-art literature in the domain of non-invasive and continuous BP monitoring with a focus on neonatal cohorts. We found that very few studies considered neonatal or paediatric cohorts, despite an active body of literature focused on an adult cohort. The neonatal cohort studies identified featured small patient cohorts and utilized foundational machine learning techniques. While results reported by these studies indicated that their proposed methods were not yet suitable for clinical use, they showed evidence that heart activity data obtained via non-invasive sensors can be linked to blood pressure. 

Meanwhile, studies focused on adult cohorts showed that strong linkages can be made between heart activity signals and BP where entire waveform segments are used to train advanced AI models, including CNN-LSTM and transformer architectures. Additionally, it has been shown that explainable artificial intelligence techniques can help to reveal key waveform features predictive of BP in the adult cohort.

Taken together, the findings of our review indicate that there is significant research opportunity in applying novel data preprocessing techniques, advanced artificial intelligence architectures, and explainable artificial intelligence tools to BP monitoring for the neonatal cohort. Research in this area could ultimately result in a significant change to clinical practice in terms of reducing risk to the patient, minimizing parental anxiety, and improving the utilization of resources, all without increasing the time burden on neonatal clinicians.

## Figures and Tables

**Figure 1 healthcare-11-03107-f001:**
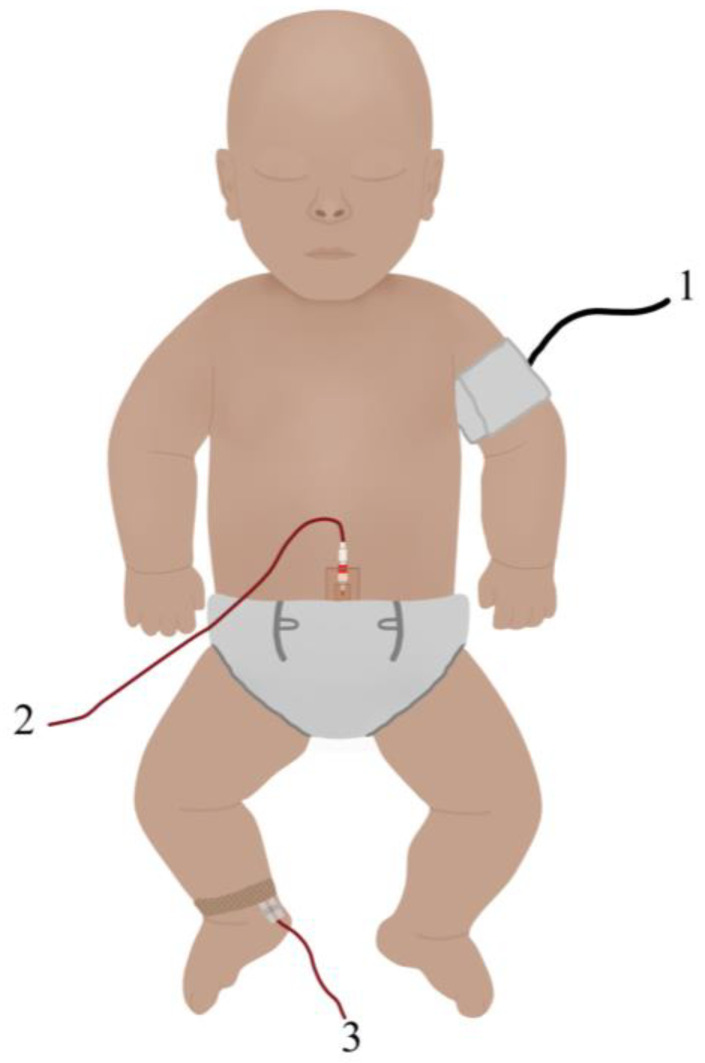
Current methods for blood pressure monitoring in neonates: (1) non-invasive cuff-based sphygmomanometer, (2) umbilical intra-arterial monitoring, and (3) peripheral intra-arterial monitoring.

**Figure 2 healthcare-11-03107-f002:**
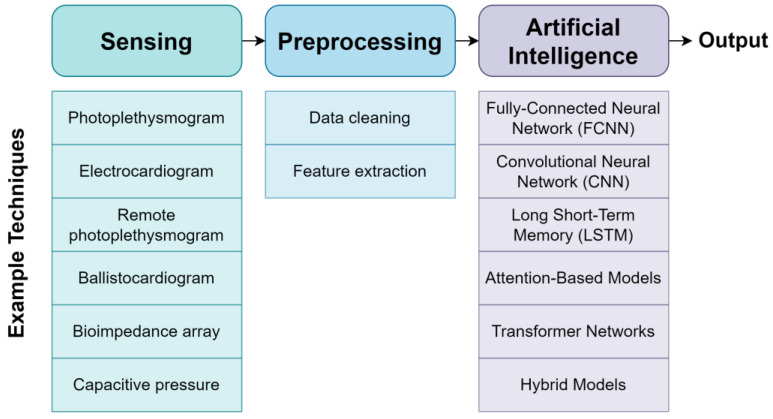
Non-invasive blood pressure monitoring pipeline with examples of possible technologies and techniques.

**Table 1 healthcare-11-03107-t001:** Grading criteria for the BHS Protocol.

Blood Pressure Monitor Grade	% Measurements with Mean Absolute Error of:		
	≤5 mmHg	≤10 mmHg	≤15 mmHg
A	60%	85%	95%
B	50%	75%	90%
C	40%	65%	80%
D	Worse than C		

## Data Availability

No new data were created or analysed in this study. Data sharing is not applicable to this article.

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
