# Peer review of "Towards Non-Invasive and Continuous Blood Pressure Monitoring in Neonatal Intensive Care Using Artificial Intelligence: A Narrative Review"

_healthcare, 2023, doi:10.3390/healthcare11243107_

Round 1

Reviewer 1 Report

Comments and Suggestions for Authors

It is a very useful scoping review addressing the non invasive monitoring of BP in neonates. 

It would be appreciated if there is a tableau synthesis all of methods in adults or children and one in neonates with the grade evaluation of Table 1. 

Author Response

Dear reviewer 1,

We have addressed your comment as follows.

Reviewer Comment: It is a very useful scoping review addressing the non invasive monitoring of BP in neonates. It would be appreciated if there is a tableau synthesis all of methods in adults or children and one in neonates with the grade evaluation of Table 1. 

Author Response: Thank you for this suggestion and for your positive feedback on our manuscript. While we agree that a comparison table would be useful, unfortunately the BHS grade is not reported by all papers. Additionally, the BHS protocol has only been validated for adult cohorts at this stage.

Thank you for taking the time to review our manuscript and provide feedback.

Kind regards,

The authors

Reviewer 2 Report

Comments and Suggestions for Authors

The manuscript effectively highlights the critical need for non-invasive continuous blood pressure monitoring in neonatal care and the potential role of AI and sensor technologies. The manuscript also highlights the critical gap in research regarding neonatal BP monitoring, where few studies have been conducted, and most of them suffer from limitations due to small sample sizes and basic AI techniques. The authors suggest several opportunities for future research, including the exploration of advanced AI architectures, explainable AI techniques, and innovative sensor technologies.

The manuscript is well-structured and logically organized. It provides a clear flow from the introduction to the literature review, methodology, results, and conclusion. However, some sections, such as those discussing specific AI models and their application in neonatal cohorts, could be more detailed. The inclusion of relevant figures or diagrams, particularly when discussing sensor technologies and AI models, would enhance the understanding of the content.

The title effectively conveys the emphasis on non-invasive blood pressure monitoring and artificial intelligence. The title's length could be reduced to enhance reader-friendliness and memorability. Think about making it more concise for a greater impact. However, the term "A Survey" lacks specificity. Readers might find it more helpful if it provides a clearer indication of the content, such as "A Review" or "Challenges and Opportunities."

Line 35: When mentioning that approximately 20% of premature neonates develop hypotension in the first 24 hours of life, it would be beneficial to provide a reference for this statistic to support the claim.

Line 66: Abbreviate "AI" the first time it is mentioned for clarity.

Materials and Methods: Consider including a PRISMA flow chart to enhance transparency. It will help readers understand the number of articles reviewed, how many were excluded, and how many were included in the review.

Line 160: The statement, "The reported error was significant and unsuitable for clinical use; however, the approach is interesting given it is entirely non-contact and thus warrants further investigation" might need rephrasing. The author's statement seems unreasonable if existing literature and research have demonstrated the unsuitability of the technique for clinical use with a high reported error.

Line 164: Enhance the elucidation of the capacitance measurement mechanism, potentially by referencing Rao et al. for a more comprehensive explanation. Consider supplementing the text with an illustrative figure to enhance clarity.

The discussion of the capacitance method seems superficial. Emphasize the potential and hopes for future research on this technique as a reasonable alternative.

Line 169: Consider mentioning research papers on ballistocardiograms in neonates used for other purposes (e.g., Joshi R, Bierling BL, Long X, et al. A Ballistographic Approach for Continuous and Non-Obtrusive Monitoring of Movement in Neonates. IEEE J Transl Eng Health Med. 2018;6:2700809). This will add depth to the discussion.

Line 188: When stating that "several other approaches offer advantages," specify the names of these approaches to provide clarity to the reader.

Line 303: When discussing Bland-Altman plots, reference the literature or source that explains how they show the level of agreement between a novel measurement technique and an existing gold standard.

Line 324: The passage regarding most research focusing on adult cohorts is repetitive and could be condensed for brevity.

Comments on the Quality of English Language

The article's quality of English language is generally good, with clear and coherent writing. 

Author Response

Dear reviewer 2,

Thank you for taking the time to review our paper and provide insightful feedback. We have addressed your comments point-by-point below. Where relevant, we have indicated new passages in our manuscript using yellow highlighting. All references are numbered consistently with the manuscript.

---

Reviewer Comment 1: The title effectively conveys the emphasis on non-invasive blood pressure monitoring and artificial intelligence. The title's length could be reduced to enhance reader-friendliness and memorability. Think about making it more concise for a greater impact. However, the term "A Survey" lacks specificity. Readers might find it more helpful if it provides a clearer indication of the content, such as "A Review" or "Challenges and Opportunities."

Author Response 1: Thank you for this suggestion. We have revised the title as follows: “Towards Non-Invasive and Continuous Blood Pressure Monitoring in Neonatal Intensive Care using Artificial Intelligence: A Narrative Review”. This is slightly shorter than our previous title, and uses the more specific “narrative review” to replace “survey”.

---

Reviewer Comment 2: Line 35: When mentioning that approximately 20% of premature neonates develop hypotension in the first 24 hours of life, it would be beneficial to provide a reference for this statistic to support the claim.

Author Response 2: Thank you for noticing the missing reference. We have added this reference to the revised manuscript, as reference [6]:

[6] S. J. Dasgupta and A. B. Gill, “Hypotension in the very low birthweight infant: the old, the new, and the uncertain.,” Arch. Dis. Child. Fetal Neonatal Ed., vol. 88, no. 6, pp. F450-4, Nov. 2003, doi: 10.1136/fn.88.6.f450.

---

Reviewer Comment 3: Line 66: Abbreviate "AI" the first time it is mentioned for clarity.

Author Response 3: Thank you for bringing our attention to this mistake. This has been corrected in the updated manuscript. The sentence in question now reads as follows:

“Several recent studies have sought to utilize artificial intelligence (AI) algorithms to estimate BP from heart activity waveforms obtained using sensors such as photoplethysmogram (PPG) and electrocardiogram (ECG) [13]-[18].”

---

Reviewer Comment 4: Materials and Methods: Consider including a PRISMA flow chart to enhance transparency. It will help readers understand the number of articles reviewed, how many were excluded, and how many were included in the review.

Author Response 4: Thank you for this suggestion. As this was a narrative review, the PRISMA method was not utilised. As such, we have not included a PRISMA diagram in our review. The primary reason for choosing a narrative review was the lack of literature pertaining to blood pressure monitoring in the neonatal cohort. We included all relevant studies on non-invasive and continuous neonatal blood pressure monitoring before expanding our search to adult cohorts in order to identify more sensors and artificial intelligence techniques that may be suitable for the neonatal cohort of interest.

To clarify that our review is a narrative review, we have made the following changes to the manuscript:

  • Our title has been revised to replace “A Survey” with “A Narrative Review”.
  • On lines 76 and 88, we have changed the phrase “exploratory review” to “narrative review” for clarity.
  • The first paragraph of Section 2 “Materials and Methods” has been expanded to include greater detail, as follows: “Our narrative review was conducted by first searching PubMed, Scopus, and Web of Science for relevant studies. Our keywords included “non-invasive”, “cuffless”, “wearable”, “blood pressure”, “prematur*”, “preterm”, “neonat*”, “baby”, “infant”, “pediatric”, “paediatric”, “child*”, “artificial intelligence”, “neural network”, “deep learning”, and “machine learning”. Studies that focused on non-invasive measurement of blood pressure or the use of AI to predict blood pressure in any age cohort were considered, with papers focused on the neonatal cohort included by default. Non-English papers were also excluded. As the field of artificial intelligence is rapidly evolving with ever-improving computational resources, we focused our search on pa-pers published between 2017-2023. Due to the large volume of papers in the adult cohort, we selected only papers published in reputable journals that utilised technologies which may be suitable for the neonatal cohort.”
  • The first sentence of Section 3 “Results” has been updated as follows: "Our exploratory search revealed that research on non-invasive BP measurement for neonatal patients, particularly those born very preterm, is limited."
  • On line 462, we have changed the phrase “scoping review” to “scoping narrative review” for clarity.

---

Reviewer Comment 5: Line 160: The statement, "The reported error was significant and unsuitable for clinical use; however, the approach is interesting given it is entirely non-contact and thus warrants further investigation" might need rephrasing. The author's statement seems unreasonable if existing literature and research have demonstrated the unsuitability of the technique for clinical use with a high reported error.

Author Response 5: Thank you for this suggestion. We have revised the sentence in question as follows for clarity:

“The reported error was significant and thus currently unsuitable for clinical use; how-ever, the approach is interesting given it is entirely non-contact. Further investigation may be warranted to improve the accuracy of this method.”

---

Reviewer Comment 6: Line 164: Enhance the elucidation of the capacitance measurement mechanism, potentially by referencing Rao et al. for a more comprehensive explanation. Consider supplementing the text with an illustrative figure to enhance clarity.

Author Response 6: Thank you for this suggestion. We have expanded our discussion of the capacitive sensing mechanism as follows:

“One work focused on a neonatal cohort used a capacitive pressure sensing array [29] built into a wearable wrist band. The capacitive sensor array is designed to measure displacement with high sensitivity. Two electrodes are built into a soft wearable band, with one electrode designed to sit against the wrist and move with the pulse while the other electrode remains relatively fixed. The small displacement changes caused by the pulse are measured as a change in capacitance, which in turn enables pulse waveforms to be derived.”

---

Reviewer Comment 7: The discussion of the capacitance method seems superficial. Emphasize the potential and hopes for future research on this technique as a reasonable alternative.

Author Response 7: Thank you for your suggestion. We have added discussion of the possible advantages and disadvantages of capacitive sensing to the end of the paragraph in question:

As this approach is not optical, it is likely to be unaffected by skin tone – thus, it is likely to work equally well on a broad range of infants, however this would need to be validated. A potential disadvantage is that its high sensitivity to displacement may lead to increased motion artifacts compared to other methods. This sensing method offers a promising direction for future research.”

---

Reviewer Comment 8: Line 169: Consider mentioning research papers on ballistocardiograms in neonates used for other purposes (e.g., Joshi R, Bierling BL, Long X, et al. A Ballistographic Approach for Continuous and Non-Obtrusive Monitoring of Movement in Neonates. IEEE J Transl Eng Health Med. 2018;6:2700809). This will add depth to the discussion.

Author Response 8: Thank you for suggesting a relevant paper. We have added the suggested work as reference [30] to our discussion of BCG sensing, as follows:

BCG sensors have also previously been utilised in the neonatal cohort for other applications including motion detection [30], and as such BP sensing could be combined with other essential monitoring using BCG.

---

Reviewer Comment 9: Line 188: When stating that "several other approaches offer advantages," specify the names of these approaches to provide clarity to the reader.

Author Response 8: Thank you for this suggestion. We have expanded the sentence in question to provide more clarity.

However, approaches discussed in this section offer advantages such as lower contact (e.g. rPPG) or more consistent performance in diverse cohorts (e.g. capacitive sensing); as such, these alternatives warrant further exploration and efforts to improve affordability.”

---

Reviewer Comment 10: Line 303: When discussing Bland-Altman plots, reference the literature or source that explains how they show the level of agreement between a novel measurement technique and an existing gold standard.

Author Response 10: Thank you for this suggestion. We have added a reference to the highly-cited Bland-Altman Lancet paper that comprehensively introduces the Bland-Altman plotting technique to the following sentence:

“Bland-Altman plots show the level of agreement between a novel measurement technique and an existing gold-standard [33].”

Reference [33] appears in the bibliography as follows:

J. M. Bland and D. G. Altman, “Statistical methods for assessing agreement between two methods of clinical measurement.,” Lancet (London, England), vol. 1, no. 8476, pp. 307–310, Feb. 1986.

---

Reviewer Comment 11: Line 324: The passage regarding most research focusing on adult cohorts is repetitive and could be condensed for brevity.

Author Response 11: Thank you for this observation. Upon reviewing the sentences in question, we decided that they were not necessary and have removed them from the revised manuscript.

---

Thank you again for taking the time to provide thorough feedback on our work. We trust that our careful revisions have addressed each of your comments and concerns.

Kind regards,

The authors

Reviewer 3 Report

Comments and Suggestions for Authors

Dear Editors,

Thank you for the opportunity to review this article. The authors have done an interesting job combining relevant clinical aspects of preterm infants with artificial intelligence. Technology in medicine, especially pediatrics, should be analyzed rigorously to ensure its potential use is based on scientific evidence. The authors' initiative is positive, and the text may be engaging. However, I believe some aspects need revision:

Although I applaud the authors' ability to summarize, some parts could be revised to be shorter. For example, the first sentence in the abstract is superfluous since practically any reader who approaches this text knows what a preterm is. Moreover, in the second sentence, it might be enough to say either the figure of fifteen million or the percentage. Offering both data seems redundant for an abstract.

Within the abstract, in the sentence "In this review, we that," some words seem missing.

In the abstract, the authors could explain the methodology better—only 1-2 lines, giving a much clearer idea of the work done.

The authors could provide some numerical results, at least the most critical ones.

Although I understand the exploratory nature of the review, the methods section needs to be developed more. The review does not seem systematized, and all scientific work must be reproducible to be refuted. Therefore, if the authors have followed a systematic approach, I recommend they describe it step by step so that any reader could reproduce it to compare the results or repeat it in another time frame. If a systematic review has not been followed, and the review has been informal or narrative, it should also be specified.

In methods, objectives are defined, which I think would fit better at the end of the introduction.

In results, it would be interesting for the authors to show quantitative data, such as the number of results obtained or a flow diagram if they have followed a systematic approach that they have been able to describe in the methods section. Regardless of the systematics, they can provide data on the articles found, discarded, analyzed, or others so that the reader can evaluate potential biases or the validity of the results.

If they showed some quantitative results that were more analytical or that could be tabulated, the text that the authors propose as results could be moved to the discussion section. The results put forward by the authors in the article seem more like a discussion than a statement of findings.

It would be helpful to add the potential limitations of the study performed. This study has limitations, such as the fact that the review could not be systematic or that there is no quantitative presentation of the results but rather a direct qualitative analysis of the studies reviewed. Whether narrative or qualitative, the limitations should be analyzed in any scientific work so the reader can better contextualize the conclusions.

Finally, the authors describe in their conclusions that they have carried out a scoping review, which should be mentioned in the manuscript. If the work performed is a scoping review, the systematics followed, and the quantitative results obtained should be presented.

In summary, the work presented by the authors is original and courageous and can help future developers. However, the article reflecting this work could be written more clearly, defining aspects such as methodology, objective results, and analyzing potential biases.

Author Response

Dear reviewer 3,

Thank you for taking the time to review our manuscript and provide detailed feedback. We have addressed each of your comments below. Where relevant, sections that have been edited in the manuscript are shown here in yellow highlighting. All references are numbered consistently with the manuscript.

---

Reviewer Comment 1: Although I applaud the authors' ability to summarize, some parts could be revised to be shorter. For example, the first sentence in the abstract is superfluous since practically any reader who approaches this text knows what a preterm is. Moreover, in the second sentence, it might be enough to say either the figure of fifteen million or the percentage. Offering both data seems redundant for an abstract.

Author Response 1: Thank you for this feedback. While we agree that most readers approaching this text from a healthcare background will know what preterm birth is, readers approaching this text from a machine learning background may not. As such, we have kept the first sentence defining preterm birth the same.

In accordance with your suggestion, we have shortened the second sentence to include only the percentage statistic. The updated sentence now reads as follows:

Approximately 11% of babies are born preterm annually worldwide.”

---

Reviewer Comment 2: Within the abstract, in the sentence "In this review, we that," some words seem missing.

Author Response 2: Thank you for noting this error. We have updated the sentence as follows:

In this review, we find that artificial intelligence is a promising tool for the continuous measurement of BP in a neonatal cohort, based on data obtained from non-invasive sensors.

---

Reviewer Comment 3: In the abstract, the authors could explain the methodology better—only 1-2 lines, giving a much clearer idea of the work done.

Author Response 3: Thank you for this suggestion. We have modified the abstract for clarity. The key sentences pertaining to the methodology and findings are as follows:

In this narrative review, we find that artificial intelligence (AI) is a promising tool for the continuous measurement of BP in a neonatal cohort, based on data obtained from non-invasive sensors. Our findings highlight key sensing technologies, AI techniques, and model assessment metrics for BP sensing in the neonatal cohort. Moreover, our findings show that non-invasive BP monitoring leveraging AI has shown promise in adult cohorts but has not been broadly explored for neonatal cohorts.”

---

Reviewer Comment 4: The authors could provide some numerical results, at least the most critical ones.

Author Response 4: Thank you for this suggestion, however as this is a narrative review, we have no specific numerical data to share.

---

Reviewer Comment 5: Although I understand the exploratory nature of the review, the methods section needs to be developed more. The review does not seem systematized, and all scientific work must be reproducible to be refuted. Therefore, if the authors have followed a systematic approach, I recommend they describe it step by step so that any reader could reproduce it to compare the results or repeat it in another time frame. If a systematic review has not been followed, and the review has been informal or narrative, it should also be specified.

Author Response 5: Thank you for this comment. Our review was a narrative review, not a systematic review. This decision was made due to the lack of literature pertaining to blood pressure monitoring in the neonatal cohort. We included all relevant studies on non-invasive and continuous neonatal blood pressure monitoring before expanding our search to adult cohorts in order to identify more sensors and artificial intelligence techniques that may be suitable for the neonatal cohort of interest.

To clarify that our review is a narrative review, we have made the following changes to the manuscript:

  • Our title has been revised to replace “A Survey” with “A Narrative Review”.
  • On lines 76 and 88, we have changed the phrase “exploratory review” to “narrative review” for clarity.
  • The first paragraph of Section 2 “Materials and Methods” has been expanded to include greater detail, as follows: “Our narrative review was conducted by first searching PubMed, Scopus, and Web of Science for relevant studies. Our keywords included “non-invasive”, “cuffless”, “wearable”, “blood pressure”, “prematur*”, “preterm”, “neonat*”, “baby”, “infant”, “pediatric”, “paediatric”, “child*”, “artificial intelligence”, “neural network”, “deep learning”, and “machine learning”. Studies that focused on non-invasive measurement of blood pressure or the use of AI to predict blood pressure in any age cohort were considered, with papers focused on the neonatal cohort included by default. Non-English papers were also excluded. As the field of artificial intelligence is rapidly evolving with ever-improving computational resources, we focused our search on pa-pers published between 2017-2023. Due to the large volume of papers in the adult cohort, we selected only papers published in reputable journals that utilised technologies which may be suitable for the neonatal cohort.”
  • The first sentence of Section 3 “Results” has been updated as follows: Our exploratory search revealed that research on non-invasive BP measurement for neonatal patients, particularly those born very preterm, is limited.
  • On line 462, we have changed the phrase “scoping review” to “scoping narrative review” for clarity.

---

Reviewer Comment 6: In methods, objectives are defined, which I think would fit better at the end of the introduction.

Author Response 6: Thank you for this suggestion. The key outcomes are mentioned in the introduction, however the objectives are reiterated in the methods due to these being the key outcomes of the initial literature search. For clarity, we have expanded the first sentence of the paragraph in question as follows:

“The first outcome of this literature search was to identify several non-invasive sensors that can be used to obtain cardiac activity waveforms on a continuous basis.”

These objectives are also discussed in the introduction as suggested. For easy reference, the paragraph included in the introduction is as follows:

“We then explore key technologies, including non-invasive sensors and artificial intelligence, which are critical to the development of non-invasive BP monitoring tools. Additionally, we introduce key metrics for assessing the performance of BP monitoring tools. As most identified works are focused on an adult cohort, we then place a lens on the few pioneering works which have considered the neonatal cohort.”

---

Reviewer Comment 7: In results, it would be interesting for the authors to show quantitative data, such as the number of results obtained or a flow diagram if they have followed a systematic approach that they have been able to describe in the methods section. Regardless of the systematics, they can provide data on the articles found, discarded, analyzed, or others so that the reader can evaluate potential biases or the validity of the results.

Author Response 7: Thank you for your comment. To clarify, this is a scoping narrative review due to paucity of neonatal data. While all papers pertaining to non-invasive and continuous blood pressure monitoring in neonates, there was a clear lack of research in the neonatal cohort, leading us to include several high-quality papers focused on the adult cohort that used technologies which may be suitable in neonatal medicine. Overall, a key finding of our paper was that there is little research in the neonatal cohort for non-invasive blood pressure monitoring, and thus our paper proposes further research in this area.

We have made several changes to our manuscript, mentioned in our response to Comment 5 above in order to clarify the narrative nature of our review.

---

Reviewer Comment 8: If they showed some quantitative results that were more analytical or that could be tabulated, the text that the authors propose as results could be moved to the discussion section. The results put forward by the authors in the article seem more like a discussion than a statement of findings.

Author Response 8: Thank you for your comment. As this is a scoping/narrative review, we don’t have any quantitative data to present in this manuscript. The information provided is more representative of our views based on our interpretation of the available literature that we reviewed. We plan to carry out research in this area and will publish our findings in due course.

We have made several changes to our manuscript, mentioned in our response to Comment 5, in order to clarify the narrative nature of our review.

---

Reviewer Comment 9: It would be helpful to add the potential limitations of the study performed. This study has limitations, such as the fact that the review could not be systematic or that there is no quantitative presentation of the results but rather a direct qualitative analysis of the studies reviewed. Whether narrative or qualitative, the limitations should be analyzed in any scientific work so the reader can better contextualize the conclusions.

Author Response 8: Thank you for your comment. In response to your suggestion, we have added a new section “Section 3.3. Limitations”, which reads as follows:

3.3. Limitations

This work presents a comprehensive narrative review of non-invasive and continuous blood pressure monitoring in the neonatal cohort. Every effort has been made to include all papers focused on this topic. Due to the relatively small body of work in this domain, our narrative review was expanded to include papers focused on the adult cohort which utilized technologies that are likely to be candidates for use in the neonatal cohort. The key limitation of this paper is that the review is narrative, not systematic, and thus there may be additional papers in the adult cohort which are not included here despite being relevant to the topic. However, there are many reviews focusing on non-invasive blood pressure monitoring in other cohorts, for example the recent works by Maqsood, et al. [40] and Zhao, et al. [41].

The new references included in this section are as follows:

[40]    S. Maqsood et al., “A survey: From shallow to deep machine learning approaches for blood pressure estimation using biosensors,” Expert Syst. Appl., vol. 197, p. 116788, 2022, doi: https://doi.org/10.1016/j.eswa.2022.116788.

[41]    L. Zhao et al., “Emerging sensing and modeling technologies for wearable and cuffless blood pressure monitoring,” npj Digit. Med., vol. 6, no. 1, p. 93, 2023, doi: 10.1038/s41746-023-00835-6.

---

Reviewer Comment 10: Finally, the authors describe in their conclusions that they have carried out a scoping review, which should be mentioned in the manuscript. If the work performed is a scoping review, the systematics followed, and the quantitative results obtained should be presented.

Author Response 10: Thank you for this comment. Our work is a scoping narrative review. We have made several changes to our manuscript, mentioned in our response to Comment 5, in order to clarify the narrative nature of our review.

---

Reviewer Comment 11: In summary, the work presented by the authors is original and courageous and can help future developers. However, the article reflecting this work could be written more clearly, defining aspects such as methodology, objective results, and analyzing potential biases.

Author Response 11: Thank you for your detailed feedback on our manuscript. We believe that our responses to your previous comments have addressed the key concerns.

---

Thank you again for taking the time to provide insightful feedback on our manuscript. We that our careful revision has addressed each of your comments and questions.

Kind regards,

The authors.